# Homogeneity in the association of body mass index with type 2 diabetes across the UK Biobank: A Mendelian randomization study

Michael Wainberg[1], Anubha Mahajan[2,3], Anshul Kundaje[1,4], Mark I. McCarthy[2,3,5¤], Erik Ingelsson[6,7,8,9], Nasa Sinnott-Armstrong[4]*, Manuel A. Rivas[10]*

1 Department of Computer Science, Stanford University, Stanford, California, United States of America, 2 Wellcome Centre for Human Genetics, University of Oxford, Oxford, United Kingdom, 3 Oxford Centre for Diabetes, Endocrinology and Metabolism, University of Oxford, Churchill Hospital, Oxford, United Kingdom, 4 Department of Genetics, Stanford University, Stanford, California, United States of America, 5 NIHR Oxford Biomedical Research Centre, Churchill Hospital, Oxford, United Kingdom, 6 Molecular Epidemiology, Department of Medical Sciences, Uppsala University, Uppsala, Sweden, 7 Science for Life Laboratory, Uppsala University, Uppsala, Sweden, 8 Division of Cardiovascular Medicine, Department of Medicine, Stanford University School of Medicine, Stanford, California, United States of America, 9 Stanford Cardiovascular Institute, Stanford University, Stanford, California, United States of America, 10 Department of Biomedical Data Science, Stanford University, Stanford, California, United States of America

¤ Current address: OMNI Human Genetics, Genentech, South San Francisco, California, United States of America

* nasa@stanford.edu (NSA); mrivas@stanford.edu (MAR)

## Abstract

### Background

Lifestyle interventions to reduce body mass index (BMI) are critical public health strategies for type 2 diabetes prevention. While weight loss interventions have shown demonstrable benefit for high-risk and prediabetic individuals, we aimed to determine whether the same benefits apply to those at lower risk.

### Methods and findings

We performed a multi-stratum Mendelian randomization study of the effect size of BMI on diabetes odds in 287,394 unrelated individuals of self-reported white British ancestry in the UK Biobank, who were recruited from across the United Kingdom from 2006 to 2010 when they were between the ages of 40 and 69 years. Individuals were stratified on the following diabetes risk factors: BMI, diabetes family history, and genome-wide diabetes polygenic risk score. The main outcome measure was the odds ratio of diabetes per 1-kg/m$^2$ BMI reduction, in the full cohort and in each stratum. Diabetes prevalence increased sharply with BMI, family history of diabetes, and genetic risk. Conversely, predicted risk reduction from weight loss was strikingly similar across BMI and genetic risk categories. Weight loss was predicted to substantially reduce diabetes odds even among lower-risk individuals: for instance, a 1-kg/m$^2$ BMI reduction was associated with a 1.37-fold reduction (95% CI 1.12–1.68) in diabetes odds among non-overweight individuals (BMI < 25 kg/m$^2$) without a family history of diabetes, similar to that in obese individuals (BMI ≥ 30 kg/m$^2$) with a family history (1.21-fold

**Data Availability Statement:** The UK Biobank data underlying the results presented in this study are available from the UK Biobank

(https://www.ukbiobank.ac.uk) for researchers meeting the criteria for data access.

**Funding:** This research has been conducted using the UK Biobank Resource under Application Number 24983, "Generating effective therapeutic hypotheses from genomic and hospital linkage data" (http://www.ukbiobank.ac.uk/wp-content/uploads/2017/06/24983-Dr-Manuel-Rivas.pdf). M. I.M. is a Wellcome and NIHR senior investigator. This work was funded in part by the Natural Sciences and Engineering Research Council of Canada (NSERC) (grant PGSD3-476082-2015 to M.W.), Stanford Bio-X Bowes fellowship (to M.W.), Stanford Graduate Fellowship (to N.S.-A.), National Defense Science & Engineering Grant (to N.S.-A.), NIH grants 1U24HG008956, R01HG010140 and 5U01HG009080 (to M.A.R.), 1DP2OD022870 and U01HG009431 (to A.K.) and 1R01DK106236, 1R01HL135313 and 1P30DK116074-01 (to E.I.), and Wellcome (090532, 098381, 203141), NIDDK (U01-DK105535) and NIHR (NF-SI-0617-10090) grants (to M.I.M.). The views expressed in this article are those of the authors and not necessarily those of the funders; funders had no role in study design, data collection and analysis, decision to publish, or preparation of the manuscript.

**Competing interests:** I have read the journal's policy and the authors of this manuscript have the following competing interests: M.I.M has served on advisory panels for Pfizer, NovoNordisk, Zoe Global; received honoraria from Merck, Pfizer, NovoNordisk and Eli Lilly; has stock options in Zoe Global; and has received research funding from Abbvie, Astra Zeneca, Boehringer Ingelheim, Eli Lilly, Janssen, Merck, NovoNordisk, Pfizer, Roche, Sanofi Aventis, Servier, Takeda. As of June 2019, M.I.M is an employee of Genentech and a holder of stock in Roche.

**Abbreviations:** BMI, body mass index; CAD, coronary artery disease; MR, Mendelian randomization.

reduction, 95% CI 1.13–1.29). A key limitation of this analysis is that the BMI-altering DNA sequence polymorphisms it studies represent cumulative predisposition over an individual's entire lifetime, and may consequently incorrectly estimate the risk modification potential of weight loss interventions later in life.

## Conclusions

In a population-scale cohort, lower BMI was consistently associated with reduced diabetes risk across BMI, family history, and genetic risk categories, suggesting all individuals can substantially reduce their diabetes risk through weight loss. Our results support the broad deployment of weight loss interventions to individuals at all levels of diabetes risk.

## Author summary

### Why was this study done?

- Excessive body weight is a key risk factor for type 2 diabetes, and weight loss is known to dramatically reduce risk, at least among people who were at high risk to begin with.

- However, even people without obvious risk factors like excessive weight or a family history of the disease still have a relatively large chance (about 1 percent) of developing type 2 diabetes: Could these individuals also reduce their risk of type 2 diabetes through weight loss?

### What did the researchers do and find?

- We looked at inherited genetic mutations that predispose people to lower body weight, and asked how much these mutations tend to protect people from type 2 diabetes, across 287,394 self-reported white British individuals from the UK Biobank cohort.

- We found that these mutations seem to offer about the same degree of protection against type 2 diabetes regardless of a person's body weight, family history of type 2 diabetes, or genetic risk for type 2 diabetes, suggesting that weight loss would have a similarly uniform protective effect for all individuals.

### What do these findings mean?

- These findings suggest that all individuals can substantially reduce their type 2 diabetes risk through weight loss, and support the broad deployment of weight loss interventions to individuals at all levels of diabetes risk as a public health measure.

- However, a key limitation to keep in mind is that genetic mutations, because they act across an individual's entire lifespan, are not a perfect proxy for weight loss interventions that happen only later in life.

## Introduction

Type 2 diabetes is a chronic metabolic condition characterized by insulin resistance and impaired insulin secretion [1], and is a leading cause of disability and death globally, partly due to cardiovascular complications [2]. Obesity is a primary risk factor for type 2 diabetes [3,4], and lifestyle interventions including weight loss and exercise reduced diabetes risk by 58% in a randomized clinical trial among overweight, prediabetic individuals with elevated post-load and fasting plasma glucose levels [5]; similar results have been observed in earlier studies [6,7]. Observational studies have also provided evidence that individuals overweight during childhood but not adulthood have reduced incidence of diabetes compared to persistently overweight individuals [8], and that overweight and obese individuals who actively participate in weight loss lifestyle change programs experience lower subsequent incidence of diabetes [9]. Interventions to reduce obesity will become increasingly relevant to public health as obesity rates worldwide continue to rise [10].

Despite the clear benefit of weight loss interventions in reducing diabetes among high-risk individuals, it is unclear whether these results generalize to lower-risk individuals. For instance, it is unclear whether weight loss causes individuals who have a normal weight to begin with, and lack genetic or familial risk factors, to reduce their diabetes risk even further. There might be a "plateau" beyond which additional weight loss has only a minor effect on diabetes risk.

More generally, other factors such as genetics and family history could modulate the effectiveness of weight loss on risk reduction. Genetics is a known driver of response to pharmaceutical interventions in general [11] and to metformin, the most commonly prescribed diabetes therapeutic, in particular [12]; it stands to reason that the same may be true for non-pharmaceutical interventions such as weight loss. On the other hand, genetic risk does not appear to modulate the risk-reducing effects of healthy lifestyle factors on coronary artery disease (CAD), with favorable genetics and lifestyle appearing to act independently of each other to reduce risk [13]. Understanding the dynamics of this process—which factors do and do not influence how weight loss translates into reduced diabetes risk—could provide insights into disease pathophysiology and inform personalized medicine.

In this study, we consider 3 factors that could plausibly modulate the risk reduction effectiveness of weight loss: initial body mass index (BMI, defined as weight in kilograms divided by squared height in meters), family history (whether a mother, father, or sibling has diabetes), and genetic risk according to a polygenic risk score constructed from over 100,000 DNA sequence polymorphisms based on their association with type 2 diabetes in a recent genome-wide association study [14]. We explore how these factors may influence the effectiveness of weight loss interventions on diabetes risk reduction in a population-scale cohort, the UK Biobank [15], encompassing over 280,000 individuals of self-reported white British ancestry with medical histories and genome-wide DNA sequence polymorphism data.

## Methods

### Study population

The UK Biobank is a prospective cohort comprising over 500,000 British individuals recruited in 2006–2010 at age between 40 and 69 years. Our analysis was restricted to 287,394 unrelated individuals of self-reported white British ancestry with anthropometrically measured BMI, type 2 diabetes diagnosis status, and self-reported ascertainment of family history of diabetes. Self-reported white British ancestry was provided as the "in_white_British_ancestry_subset" column of the UK Biobank sample quality control file (ukb_sqc_v2.txt). For quality control,

we also excluded individuals not used to compute genotype principal components ("used_in_pca_calculation" column of ukb_sqc_v2.txt), individuals flagged as outliers in heterozygosity or missingness ("het_missing_outliers" column), individuals displaying putative sex chromosome aneuploidy ("putative_sex_chromosome_aneuploidy" column), and individuals with more than 10 putative third-degree relatives ("excess_relatives" column).

Of these 287,394 individuals, 13,982 (4.9%) were considered type 2 diabetes cases and 273,412 (95.1%) controls. Type 2 diabetes was defined based on Eastwood et al. [16] using "probable type 2 diabetes" and "possible type 2 diabetes" as cases and "type 2 diabetes unlikely" as controls; we excluded individuals with "probable type 1 diabetes" or "possible type 1 diabetes" or "possible gestational diabetes" and controls with HbA1C $\geq$ 39 mmol/mol, as this is indicative of undiagnosed diabetes or prediabetes [17]. Except where noted, "diabetes" refers to type 2 diabetes throughout the text.

All individuals had medical history data, as well as DNA sequence polymorphism data for approximately 800,000 markers ascertained using the UK Biobank Axiom or UK BiLEVE genotyping array and imputed using a combination of the UK10K, 1000 Genomes, and Haplotype Reference Consortium reference panels [18]. We used the following types of data from each of these individuals: the aforementioned polymorphism data, age, sex, genotyping array used (Axiom or BiLEVE), type 2 diabetes diagnosis, self-reported family history of diabetes (mother, father, and/or sibling), BMI calculated from measured height and weight, medication status for insulin and metformin, and, to correct for residual population stratification, UK Biobank assessment center and the top 40 principal components of the DNA sequence polymorphism matrix across individuals [19].

## Polygenic risk score

We calculated the polygenic risk score for each of the individuals in our cohort using the 136,795-polymorphism score from a recent diabetes genome-wide association study [20]. The polygenic risk score was imperfectly predictive of disease status (area under the receiver operating characteristic curve [AUC] 0.66), owing both to type 2 diabetes having a strong environmental component, rather than being purely genetic, and to the substantial missing heritability [21] not yet accounted for by genome-wide association studies.

## Mendelian randomization

We stratified individuals on BMI (non-overweight, BMI < 25 kg/m$^2$, $N$ = 100,294; overweight, $25 \leq$ BMI < 30 kg/m$^2$, $N$ = 123,820; obese, BMI $\geq$ 30 kg/m$^2$, $N$ = 63,280), family history (mother, father, or any sibling with diabetes, $N$ = 48,238), and polygenic risk score for type 2 diabetes derived from genome-wide summary statistics with and without BMI adjustment (low, medium, or high tertile, $N$ = 95,798). We also stratified diabetes cases based on metformin ($N$ = 7,923 of 13,982) or insulin ($N$ = 2,094 of 13,982) prescription, while using the full set of control individuals. Within each subset of individuals, as well as in the full cohort, we performed inverse-variance-weighted Mendelian randomization (MR) and MR–Egger regression [22] and used the MR–Egger intercept test to assess the impact of horizontal pleiotropy. We refer the reader to Emdin et al. [23] for a comprehensive introduction to MR and its applications to medicine.

First, we performed genome-wide association studies for BMI and diabetes in UK Biobank with linear and logistic regression, respectively, using the PLINK software package [24] (version 2.0), restricted to only the individuals in each respective subset. We included as covariates age, sex, genotyping array, UK Biobank assessment center, and the top 40 global principal components of the DNA sequence polymorphism matrix. To mitigate bias in MR effect sizes

due to participant overlap [25], we restricted the BMI genome-wide association study to only type 2 diabetes controls.

We then performed MR using genome-wide-significant BMI-associated polymorphisms from the GIANT consortium [26] curated by a previous MR study [27] as instrumental variables. Although a recent study [28] meta-analyzed this GIANT consortium GWAS with the UK Biobank and found over 500 independent BMI-associated polymorphisms, we chose not to use this GWAS for instrument selection because MR instruments should be chosen from an external GWAS (in this case, one that does not include UK Biobank data) to avoid winner's curse.

For quality control [29], we required polymorphisms to be missing in fewer than 10% of individuals ("--geno 0.1") and to have a Hardy–Weinberg equilibrium $p$-value of greater than $1 \times 10^{-20}$ ("--hwe 1e-20 midp") [30]. Overall, 57 of 69 instruments passed these thresholds: rs7899106, rs17094222, rs11191560, rs4256980, rs2176598, rs3817334, rs12286929, rs7138803, rs11057405, rs9581854, rs12429545, rs10132280, rs12885454, rs7141420, rs3736485, rs758747, rs12446632, rs2650492, rs1558902, rs1000940, rs12940622, rs1808579, rs7243357, rs6567160, rs17724992, rs29941, rs2287019, rs657452, rs3101336, rs17024393, rs543874, rs2820292, rs13021737, rs11126666, rs1016287, rs11688816, rs2121279, rs1528435, rs7599312, rs6804842, rs3849570, rs13078960, rs16851483, rs1516725, rs10938397, rs11727676, rs2112347, rs2207139, rs13191362, rs1167827, rs17405819, rs2033732, rs4740619, rs10968576, rs6477694, rs1928295, and rs10733682. These 57 polymorphisms are conceptually different from the variants in the polygenic risk score, since they were chosen based on their association with BMI, whereas the polymorphisms in the polygenic risk score were chosen based on their association with diabetes. Furthermore, the polygenic risk score used thousands of polymorphisms to capture as much trait variance as possible, while these 57 polymorphisms were selected to have individually measurable effects on BMI.

We tested for non-linearity in the relationship between diabetes odds ratio and BMI using the method of Staley and Burgess [31]. We constructed an "allele score" as a sum of the 57 instruments, weighted by their effect sizes in Locke et al.'s smoking-adjusted European BMI summary statistics (BMI.SNPadjSMK.CombinedSexes.EuropeanOnly.txt; 1 instrument, rs9581854, did not appear in this file and was consequently excluded from the allele score). We input this allele score into Staley and Burgess's method to estimate diabetes odds ratios within 50 BMI quantiles while accounting for collider bias. We then calculated a Cochran Q $p$-value to test for heterogeneity across the 50 quantiles. We also calculated Cochran Q and trend test $p$-values (as described in [31]) for whether the association of the allele score with BMI varied across the 50 quantiles; an assumption of the method is that it does not.

### Mitigating collider bias

To mitigate collider bias [32], we ensured that all variables stratified on (BMI, family history, and polygenic risk score) were independent of the 57 instruments, using a combination of 2 strategies. For BMI and the polygenic risk score, we simply regressed out the 57 instruments (i.e., by predicting BMI from the instruments using multilinear regression, then mean-centering the predictions and subtracting them from BMI to create an adjusted BMI; and similarly for the polygenic risk score). Throughout the text, "BMI" and "polygenic risk score" refer to individuals' BMIs and polygenic risk scores after regressing out the 57 instruments. Note that since these 57 instruments collectively explain only a small proportion of variance in BMI ($r^2 = 0.013$) and polygenic risk ($r^2 = 0.003$), the magnitude of this adjustment is minimal, and the adjusted BMIs and polygenic risk scores are approximately equal to the unadjusted values.

For family history, which unlike BMI and polygenic risk is binary, we instead corrected for collider bias by predicting family history from the instruments using multivariate logistic regression, then matching individuals between the family history and no family history groups on the logistic regression predictions in a 4.5:1 ratio. Specifically, we binned individuals into 10 deciles based on the predictions and, within each decile, randomly subsampled the no family history group to 4.5 times the size of the family history group (rounded to the nearest integer number of individuals). This subsampling led to the inclusion of a total of 48,238 individuals with family history and 217,071 without in the family history analysis; 22,085 individuals were excluded.

## Results

### Prevalence of diabetes in UK Biobank

One simple way to estimate the effectiveness of weight loss in reducing diabetes risk is to tabulate how diabetes prevalence differs between subsets of individuals with different diabetes risk factors. To this end, we stratified individuals within our self-reported white British UK Biobank cohort based on BMI, and further based on family history or polygenic risk score tertile, then tabulated diabetes prevalence within each stratum (Table 1; Fig 1 and S1 Fig).

The diabetes prevalence of 1.3% among non-overweight individuals (BMI < 25 kg/m$^2$) increased to 3.7% among overweight individuals (25 ≤ BMI < 30 kg/m$^2$) and 12.8% among obese individuals (BMI > 30 kg/m$^2$), which, as UK Biobank is not a fully representative sample of the UK population, differs somewhat from 2014 Public Health England prevalence estimates of 2.4% among healthy-weight individuals, 5.2% among overweight individuals, and 12.4% among obese individuals [34]. Family history of diabetes was associated with between doubled and tripled diabetes prevalence across all BMI categories (though this increase could be overestimated due to within-category correlations between family history and BMI; see Discussion and S2 Fig).

Polygenic risk was also associated with increased prevalence: across all BMI categories, prevalence doubled to tripled from the lowest to the highest tertile of polygenic risk, similar to the doubling to tripling observed with family history. Hence, BMI, family history, and genetics all showed association with diabetes risk, as expected, likely in part due to these 3 risk factors being correlated with each other [35].

What do these results imply for weight loss interventions? Table 1 and Fig 1 show that elevated BMI is associated with increased diabetes prevalence across all categories of individuals.

**Table 1. Prevalence of diagnosed type 2 diabetes in the self-reported white British UK Biobank cohort, stratified by BMI, family history, and polygenic risk score.**

| Group | Non-overweight (BMI < 25 kg/m$^2$) | Overweight (25 ≤ BMI < 30 kg/m$^2$) | Obese (BMI ≥ 30 kg/m$^2$) |
|---|---|---|---|
| **Overall** | 1.3% | 3.7% (2.8×) | 12.8% (9.8×) |
| **Family history of diabetes** | | | |
| No | 1.0% | 2.8% (2.8×) | 10.2% (10.2×) |
| Yes | 3.1% (3.1×) | 8.4% (8.4×) | 22.2% (22.2×) |
| **Polygenic risk score** | | | |
| Low | 0.7% | 1.8% (2.6×) | 7.1% (10.1×) |
| Medium | 1.1% (1.6×) | 3.2% (4.6×) | 11.6% (16.6×) |
| High | 2.1% (3.0×) | 6.2% (8.9×) | 18.0% (25.7×) |

Relative risks with respect to the lowest-risk (top left) category in each stratification are shown in parentheses. Because prevalences within our cohort are by definition exact, we do not report confidence intervals.

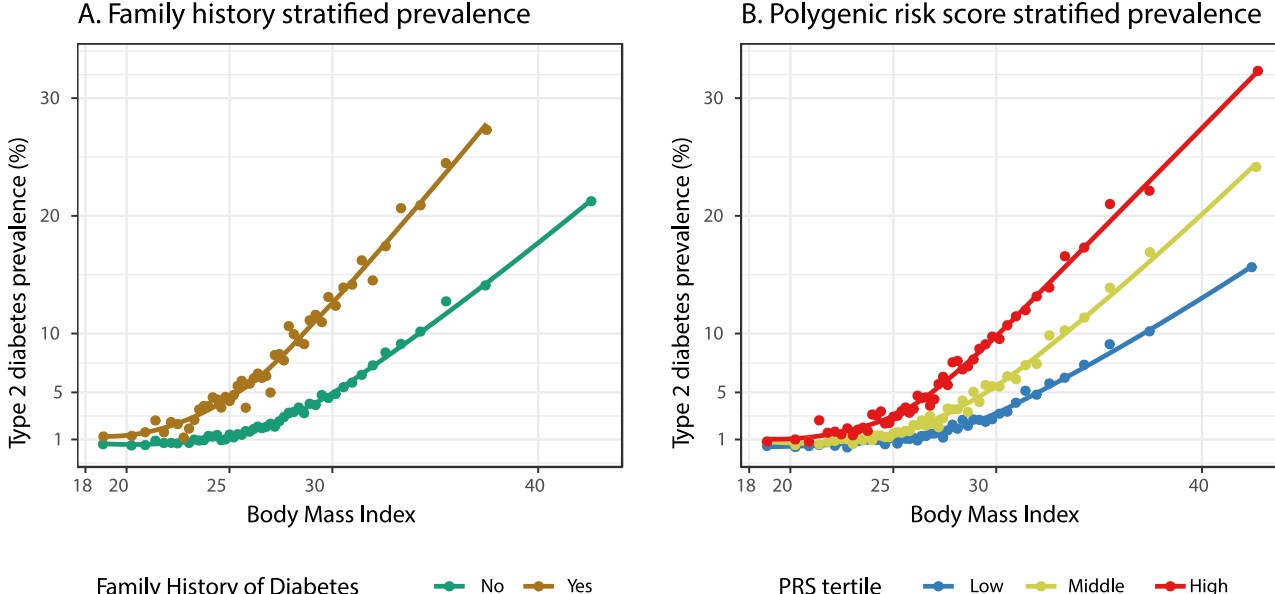

**Fig 1. Variation of type 2 diabetes prevalence with BMI, family history, and polygenic risk.** Within each category, 50 bins of equal numbers of individuals with consecutive BMI values were selected, and the average BMI and percent of individuals diagnosed with type 2 diabetes are shown. Curves were fit using LOESS regression [33]. Individuals were stratified by (A) family history of diabetes or (B) polygenic risk score (PRS) tertile.

However, we should not interpret this to suggest that all individuals can reduce their diabetes risk through weight loss, because observational studies are highly prone to confounding. One example of potential confounding in Table 1 and Fig 1 is the correlation of socioeconomic status with both BMI [36] and type 2 diabetes prevalence [37]; complicating the problem, the degree of correlation depends on how socioeconomic status is measured [36].

## MR effect size of BMI modification on diabetes risk

To mitigate some of the limitations of observational studies, we used MR [22,23], a powerful technique that aims to assess causality between traits by using DNA sequence polymorphisms as instrumental variables. MR has been likened to a quasi-randomized control trial, where individuals are randomized based on their unique set of inherited genetic polymorphisms [38], and is therefore less vulnerable to confounders than other types of observational data analysis. For instance, MR is able to rule out high-density lipoprotein cholesterol as a causal risk factor for CAD [39]. We build on a rich body of medical literature using MR in the UK Biobank dataset to interrogate epidemiological relationships [27,40–43].

Rather than reporting mere prevalences, MR is able to estimate the odds ratio of diabetes per kg/m$^2$ decrease (or increase) in BMI within each subset of individuals (Table 2 and S1 Table; S3 Fig), a critical quantity when considering weight loss interventions. Non-overweight individuals were estimated to have a 1.31-fold (95% CI 1.11–1.53) reduced odds of diabetes per unit decrease in BMI according to inverse-variance-weighted MR. This is broadly concordant with previous MR estimates of 1.26-fold, using the GIANT and DIAGRAM studies [44], and 1.19-fold, using the GIANT and Genetic Epidemiology Research on Adult Health studies and a subset of the UK Biobank cohort [45]. Overweight and obese individuals had roughly the same odds ratios as non-overweight individuals, though the odds ratio for obese individuals (1.25-fold, 95% CI 1.20–1.31) was slightly but significantly lower than that for overweight individuals (1.36-fold, 95% CI 1.28–1.45; $p$ = 0.03). There also appeared to be some

**Table 2. Diabetes odds ratio per kg/m$^2$ increase in BMI within various subsets of individuals, according to inverse-variance-weighted Mendelian randomization.**

|  | Non-overweight (BMI < 25 kg/m$^2$) | Overweight (25 ≤ BMI < 30 kg/m$^2$) | Obese (BMI ≥ 30 kg/m$^2$) |
|---|---|---|---|
| **Overall** | 1.31 (1.11, 1.53) | 1.36 (1.28, 1.45) | 1.25 (1.20, 1.31) |
| **Family history of diabetes** |  |  |  |
| No | 1.37 (1.12, 1.68) | 1.34 (1.24, 1.44) | 1.24 (1.18, 1.32) |
| Yes | 1.09 (0.87, 1.36) | 1.34 (1.21, 1.49) | 1.21 (1.13, 1.29) |
| **Polygenic risk score** |  |  |  |
| Low | 1.17 (0.82, 1.65) | 1.29 (1.11, 1.48) | 1.22 (1.12, 1.34) |
| Medium | 1.59 (1.21, 2.10) | 1.35 (1.19, 1.53) | 1.28 (1.19, 1.38) |
| High | 1.16 (0.93, 1.45) | 1.40 (1.29, 1.53) | 1.25 (1.18, 1.32) |
| **Diabetes medication** |  |  |  |
| Insulin only | 1.39 (0.98, 1.97) | 1.32 (1.12, 1.55) | 1.33 (1.22, 1.44) |
| Metformin only | 1.44 (1.12, 1.85) | 1.49 (1.37, 1.61) | 1.30 (1.23, 1.38) |

95% confidence intervals are indicated in parentheses.

heterogeneity within BMI categories (S2 Table), particularly between obese individuals with BMI < 35 kg/m$^2$ (1.38-fold, 95% CI 1.29–1.47) versus ≥35 kg/m$^2$ (1.13-fold, 95% CI 1.05–1.21; $p = 6 \times 10^{-5}$); however, using the method of Staley and Burgess [31], we did not find significant evidence of heterogeneity across 50 BMI quantiles (Cochran Q heterogeneity $p = 0.6$; S4 Fig). Polygenic risk score and family history also appeared to have minimal influence on odds ratio, with no significant differences based on family history or between PRS groups for any BMI category (S1 Table). Results were broadly concordant using MR–Egger regression instead of inverse-variance-weighted MR (S3 and S4 Tables).

We also considered the influence of diabetes medication on the results, to investigate the influence of confounding due to reverse causality of these medications on BMI [5] or differences in diabetes severity between medicated and non-medicated individuals. When we restricted our set of diabetes cases to individuals prescribed metformin (7,923 of 13,982 individuals) or insulin (2,094 of 13,982 individuals), while using the full set of control individuals, odds ratios were again similar (Table 2). On the whole, MR predicted that all individuals have a substantial ability to reduce their diabetes risk through weight loss, regardless of BMI, family history, or genetic risk.

## Relative versus absolute risk reduction

While overweight and obese individuals are predicted to have similar *relative* risk reduction with weight loss, it is also worth considering how this translates into *absolute* risk reduction (i.e., reduction in population prevalence), which is arguably more important from a public health perspective. We predict that a hypothetical 1-kg/m$^2$ BMI reduction among all the overweight individuals in our cohort would reduce diabetes prevalence from 5.2% to 3.8% (i.e., 5.2% divided by 1.36, making the approximation that relative risk is similar to odds ratio since prevalences are sufficiently low and odds ratios are sufficiently close to 1 [46]). This would result in a substantial absolute reduction in diabetes prevalence of 1.4 percentage points (95% CI 1.1–1.6) among these overweight individuals (Table 3). By a similar calculation, we predict that the same 1-kg/m$^2$ BMI reduction among the obese individuals in our cohort would reduce prevalence by an even more substantial 2.5 percentage points (95% CI 2.1–2.9). The larger absolute risk reduction from weight loss among obese individuals suggests that existing public health efforts focused on this high-risk group have not been misplaced.

**Table 3. Relative versus absolute risk reduction across BMI categories.**

| Measure | Non-overweight (BMI < 25 kg/m$^2$) | Overweight (25 ≤ BMI < 30 kg/m$^2$) | Obese (BMI ≥ 30 kg/m$^2$) |
|---|---|---|---|
| Relative odds ratio, from Mendelian randomization (β) | 1.31 (1.11, 1.53) | 1.36 (1.28, 1.45) | 1.25 (1.20, 1.31) |
| Diabetes prevalence (P), from Public Health England [34] | 2.40% | 5.20% | 12.40% |
| Prevalence after 1-kg/m$^2$ BMI reduction (P/β) | 1.8% (1.6, 2.2) | 3.8% (3.6, 4.1) | 9.9% (9.5, 10.3) |
| Absolute risk reduction: Reduction in prevalence after 1-kg/m$^2$ BMI reduction (~P − P/β) | 0.6% (0.2, 0.8) | 1.4% (1.1, 1.6) | 2.5% (2.1, 2.9) |

Absolute risk reduction for a 1-kg/m$^2$ reduction in BMI is estimated from the baseline diabetes prevalence (P) and Mendelian randomization odds ratio (β) as P − P/β. 95% confidence intervals are indicated in parentheses.

### Example: Risk reduction for a fixed amount of weight loss

Consider an individual of 1.7 meters (approximately 5 feet, 7 inches) in height. If the individual is overweight (72 kg [159 lbs] to 87 kg [192 lbs]) and loses 2.3 kg (5 lbs), MR predicts a 22% relative reduction in diabetes risk (95% CI 18%–25%), or a 0.8% absolute reduction (95% CI 0.7%–0.9%), according to the same type of calculation as in Table 3. (This analysis relies on the multiplicative nature of odds ratios, so that for instance a 2.5-kg/m$^2$ BMI reduction at an odds ratio of 1.30 would result in a predicted relative risk reduction of $1 - 1/1.30^{2.5}$, or about 48%.) On the other hand, for a 1.7-meter tall obese individual (87 kg [192 lbs] or greater), the same 2.3-kg (5-lb) weight loss would result in a 16% predicted relative risk reduction (95% CI 13%–19%) and a 2.1% absolute reduction (95% CI 1.7%–2.5%). The results of similar calculations for 4.5-kg (10-lb), 5.0-kg, 9.1-kg (20-lb), and 10.0-kg weight reductions are shown in Table 4.

### Example: Weight loss for a fixed amount of risk reduction

We can also invert the problem to predict how many pounds lost would result in a given amount of risk reduction, for the same 1.7-meter-tall individual. The results of this calculation, for a 25% or 50% relative or 1% or 2% absolute risk reduction, are shown in Table 5; we predict that losing 5 kg (11 lbs) would be sufficient to reduce relative risk by 25% and absolute risk by 1% in both overweight and obese individuals, and that losing 11 kg (about 25 lbs) would be sufficient to reduce relative risk by 50% and absolute risk by 2% in both groups.

## Discussion

In this study, we quantitatively examined the influence of BMI, family history of diabetes, and genetic risk on the association between BMI and diabetes in a population-scale cohort of over 280,000 individuals. We found that all groups, even those at low risk, show evidence of an

**Table 4. Predicted relative and absolute diabetes risk reduction resulting from various amounts of weight loss, for a 1.7-meter-tall individual.**

| Weight lost | Predicted relative percent reduced risk | | Predicted absolute percent reduced risk | |
|---|---|---|---|---|
| | Overweight (25 ≤ BMI < 30 kg/m$^2$) | Obese (BMI ≥ 30 kg/m$^2$) | Overweight (25 ≤ BMI < 30 kg/m$^2$) | Obese (BMI ≥ 30 kg/m$^2$) |
| 2.3 kg (5 lbs) | 22% (18, 25) | 16% (13, 19) | 0.8% (0.7, 0.9) | 2.1% (1.7, 2.5) |
| 4.5 kg (10 lbs) | 39% (32, 44) | 30% (25, 35) | 1.4% (1.2, 1.6) | 3.8% (3.1, 4.5) |
| 5.0 kg (11 lbs) | 42% (35, 47) | 32% (27, 38) | 1.5% (1.3, 1.8) | 4.1% (3.4, 4.8) |
| 9.1 kg (20 lbs) | 62% (54, 69) | 51% (43, 58) | 2.3% (2.0, 2.6) | 6.5% (5.5, 7.4) |
| 10.0 kg (22 lbs) | 66% (58, 72) | 54% (46, 61) | 2.5% (2.1, 2.7) | 7.0% (5.9, 7.8) |

95% confidence intervals are indicated in parentheses.

**Table 5. Predicted number of pounds/kilograms that would need to be lost to achieve various percent reductions in relative or absolute diabetes risk, for a 1.7-meter-tall individual.**

| Percent reduced diabetes risk | Predicted weight loss needed to achieve risk reduction | |
| --- | --- | --- |
| | Overweight ($25 \leq$ BMI $< 30$ kg/m$^2$) | Obese (BMI $\geq 30$ kg/m$^2$) |
| 25%, relative | 6 lbs (5, 7)/3 kg (2, 3) | 8 lbs (7, 10)/4 kg (3, 5) |
| 50%, relative | 14 lbs (12, 18)/6 kg (5, 8) | 20 lbs (16, 25)/9 kg (7, 11) |
| 1%, absolute | 6 lbs (5, 8)/3 kg (2, 4) | 2 lbs (2, 3)/1 kg (1, 1) |
| 2%, absolute | 16 lbs (13, 20)/7 kg (6, 9) | 5 lbs (4, 6)/2 kg (2, 3) |

95% confidence intervals are indicated in parentheses.

association between reduced BMI and reduced diabetes risk, suggesting that all individuals have the ability to reduce their diabetes risk through weight loss. Crucially, neither family history nor genetic risk fundamentally alter the strength of this association, suggesting that despite some degree of genetic and environmental predisposition, all individuals can still take charge of their diabetes risk through lifestyle modifications. Our results support ongoing public health campaigns to encourage weight loss for all individuals. We note, however, that there is still substantial interindividual variation in treatment adherence [47], and personalized adherence strategies remain a fruitful area for further research.

There are several limitations to our study. Our use of discrete divisions for BMI, family history, and polygenic risk, though standard among epidemiological studies of this kind [13], opens up the possibility of heterogeneity in other variables within each division. For instance, although family history is associated with a large increase in risk even within each BMI category, some of this increase could be due to within-category correlations between family history and BMI. Our individuals' self-reported family history of diabetes does not distinguish between type 1 and 2 diabetes (though type 2 diabetes accounts for 90% to 95% of cohort diabetes cases) and could also be subject to ascertainment bias, such as people only becoming aware of a family history of diabetes after being diagnosed themselves. Family history may be less informative in societies undergoing demographic transition, where the parental environment may be drastically different. BMI is an imperfect proxy for body adiposity [48], and other anthropometrics, such as waist circumference, might have overlapping effects on diabetes risk. In particular, it has been shown that waist circumference is associated with diabetes risk even after accounting for BMI [49]. In addition, our observations are of present BMI and are thus concurrently ascertained with diabetes status, not recorded at or before the time of diagnosis.

While MR is designed to account for confounding due to environmental factors, and age, sex, and population structure were explicitly corrected for, there could be residual confounding, particularly due to covariates with non-linear effects. MR is also known to be vulnerable to pleiotropy [50] (e.g., polymorphisms affecting BMI and diabetes through 2 independent biological mechanisms) and reverse causality, and stratifying on the exposure (BMI) could induce correlation between the outcome (diabetes) and instrumental variables (DNA sequence polymorphisms) [31,51], contrary to an assumption of MR. As noted previously [52], the analogy between MR and randomized control trials is imperfect because of (1) violations of non-exchangeability due to population structure, (2) study eligibility criteria being defined late in life, after conception (i.e., post-randomization), (3) potentially imperfect linkage between the instruments and the true causal genetic variants, and (4) an unclear definition of adherence to treatment.

Perhaps most importantly, the estimates reported by MR, being based on inherited DNA sequence polymorphisms, represent cumulative effects of the exposure over an individual's entire lifetime, and may consequently incorrectly estimate the risk modification potential of interventions later in life, violating an implicit assumption of gene–environment interchangeability in MR [50]. Conceptually, although MR can determine that lower lifetime BMI is protective against diabetes, this does not necessarily imply that weight loss later in life, after carrying excessive weight for decades, would have the same result. We stress the need to validate the conclusions presented here via randomized controlled trial or prospective study evidence, since even MR is not fully immune to the limitations of observational cohorts. For instance, as the UK Biobank cohort continues to age, new incidence of type 2 diabetes cases might be used in a prospective fashion to estimate age-related effects.

In conclusion, by analyzing a cohort of over 280,000 individuals, we found that across BMI, family history, and genetic risk categories, genetically predicted lower BMI is consistently associated with reduced diabetes risk. This suggests that individuals still have substantial ability to reduce diabetes risk through weight loss, regardless of genetics or family history.

## Supporting information

**S1 Fig. Variation of type 2 diabetes prevalence with BMI, stratified by sex and polygenic risk.** Within each category, 50 bins of equal numbers of individuals with consecutive BMI were selected, and the average BMI and percent of individuals diagnosed with type 2 diabetes are shown. Curves were fit using LOESS regression [33]. Individuals were stratified by (A) sex or (B) polygenic risk score quantile (0%–5%, 47.5%–52.5%, 95%–100%).
(PDF)

**S2 Fig. Difference in BMI in individuals with versus without family history of diabetes.** Individuals were stratified into controls (left) and type 2 diabetes cases (right) and further stratified by the presence (brown) or absence (green) of a family history of diabetes. There was a significant difference in mean BMI between individuals with a family history of diabetes in controls ($0.8 \text{ kg/m}^2$, $t$ test $p < 0.001$) but not in cases ($0.2 \text{ kg/m}^2$, $t$ test $p = 0.1$).
(PNG)

**S3 Fig. Diabetes odds ratio per $\text{kg/m}^2$ increase in BMI within various subsets of individuals, according to inverse-variance-weighted MR.** Error bars indicate 95% confidence intervals.
(PNG)

**S4 Fig. Diabetes odds ratios across 50 BMI quantiles, using the method of Staley and Burgess.** The average of the 50 odds ratios is shown as a gray line. We found no significant evidence of heterogeneity across the 50 quantiles (Cochran Q heterogeneity $p = 0.6$). However, the association of the allele score with BMI appeared to vary slightly across the 50 quantiles (Cochran Q heterogeneity $p = 0.01$, trend test $p = 0.7$), limiting the conclusions that can be drawn from this analysis.
(PNG)

**S1 Table. $p$-Values between all adjacent groups in Table 2, calculated via a difference-of-odds-ratios test.**
(DOC)

**S2 Table. Diabetes prevalence and odds ratio per $\text{kg/m}^2$ increase in BMI computed via MR, as in Table 2, but with more fine-grained stratifications for overweight and obese individuals.**
(DOC)

**S3 Table. The results of Table 2 using MR–Egger regression instead of inverse-variance-weighted MR.**
(DOC)

**S4 Table. MR–Egger intercept test *p*-values.**
(DOC)

## Acknowledgments

We gratefully acknowledge Johanne Marie Justesen, Tara Templin, Jonathan Pritchard, Sanjay Basu, and members of the Rivas lab for helpful discussions.

## Author Contributions

**Conceptualization:** Michael Wainberg, Nasa Sinnott-Armstrong, Manuel A. Rivas.

**Data curation:** Anubha Mahajan.

**Investigation:** Michael Wainberg, Anubha Mahajan, Anshul Kundaje, Mark I. McCarthy, Erik Ingelsson, Nasa Sinnott-Armstrong, Manuel A. Rivas.

**Methodology:** Michael Wainberg, Anubha Mahajan, Anshul Kundaje, Mark I. McCarthy, Erik Ingelsson, Nasa Sinnott-Armstrong, Manuel A. Rivas.

**Supervision:** Anshul Kundaje, Mark I. McCarthy, Erik Ingelsson, Nasa Sinnott-Armstrong, Manuel A. Rivas.

**Visualization:** Michael Wainberg, Nasa Sinnott-Armstrong.

**Writing – original draft:** Michael Wainberg.

**Writing – review & editing:** Michael Wainberg, Anubha Mahajan, Anshul Kundaje, Mark I. McCarthy, Erik Ingelsson, Nasa Sinnott-Armstrong, Manuel A. Rivas.

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
