## [Decision Letter · Decision Letter 0]

2 Sep 2019

Dear Dr. Rivas,

Thank you very much for submitting your manuscript "Homogeneity in the Effect of Body Mass Index on Type 2 Diabetes across the UK Biobank: A Mendelian Randomization Study" (PMEDICINE-D-19-02124) for consideration at PLOS Medicine. 

Your paper was evaluated by a senior editor and discussed among all the editors here. It was also discussed with an academic editor (who has some comments below) with relevant expertise, and sent to independent reviewers, including a statistical reviewer. The reviews are appended at the bottom of this email and any accompanying reviewer attachments can be seen via the link below:

[LINK]

In light of these reviews, I am afraid that we will not be able to accept the manuscript for publication in the journal in its current form, but we would like to consider a revised version that addresses the reviewers' and editors' comments. Obviously we cannot make any decision about publication until we have seen the revised manuscript and your response, and we plan to seek re-review by one or more of the reviewers. 

We expect to receive your revised manuscript by Sep 23 2019 11:59PM. Please email us (plosmedicine@plos.org) if you have any questions or concerns.

We look forward to receiving your revised manuscript. 

Sincerely,

Adya Misra PhD

Senior Editor 

PLOS Medicine

plosmedicine.org

Please avoid causal language as this study is not a trial – please remove the word ‘effect’ from the title and any other incidences in the manuscript. 

Similar line 1 of the Methods and Findings section of the abstract (causal - Mendelian Randomization (MR) study of the causal effect size of BMI on diabetes odds in 287,399 unrelated individuals of white British ancestry in the UK Biobank) and causal appears multiple times in the main text. 

Please provide p values in the abstract and elsewhere where any data is quantified with 95%Cis.

Please add a sentence as the final sentence of the Methods and Findings section of the abstract on the study’s limitations.

Please provide some summary demographic data to go into an abstract from the cohort you use in your study. 

Academic Editor comments:

My only query/concern is on the 57 variants that were used to define the instrument. This is based on a 2015 GWAS, and variants curated from a previous MR analysis from a different group. It's good to have consistency between publications, but there have been many more variants associated now. 

One reviewer commented: 

"Is the opinion of this reviewer that the instrument should be updated accordingly", and another also raised concerns. 

I would be interested in the statistical reviewer's opinion. I don't feel strongly either way, but it should probably be discussed in the paper.

Comments from the reviewers:

Reviewer #1: Review of: M. Wainberg et al. "Homogeneity in the Effect of Body Mass Index on Type 2 Diabetes across the UK Biobank: A Mendelian Randomization Study"

Submitted to PLOS Medicine 

MS # MEDICINE-D-19-02124

In this manuscript, Michael Wainberg and colleagues investigated the effect of body mass index (BMI) on type 2 diabetes (T2D) using multi-stratum Mendelian randomisation (MR) analysis in the UK Biobank. Authors showed that genetically driven BMI reduction was consistently associated with reduced T2D odds across BMI, family history, and genetic risk categories. Findings suggest that all individuals can substantially reduce their T2D risk through weight loss. 

The topic, as pointed out by the authors, is of high interest given evidence from large high-quality randomized clinical trials showing that lifestyle interventions can prevent or delay the progression to T2D by ~50% among high-risk individuals. The present study attempted to extend these observations in low-risk individuals using contemporary methodological approaches. The corroboration that body weight reduction strategies may work even among individuals at low T2D risk could offer a rational framework for practice and policy by supporting dietary or lifestyle interventions for T2D to be deployed across all gradients of the population. The manuscript is clear, concise, and very-well written.

This paper has the following strengths:

1. Focus on an interesting and relevant hypothesis

2. Well-powered study with appropriate genetic and outcome information

3. Robust statistical approach

4. Potential to inform practice and policy

5. Authors are top scientist in the field of obesity and T2D genetics 

I would like to make the following suggestions in order to help improve the overall quality of the manuscript and strengthen the impact of main findings. 

BMI is instrumented by the use of 57 BMI increasing-risk variants that passed the quality control threshold defined by the authors. Weights were based on Locke at al., data. There are now more than 530 independent loci associated with BMI (Yengo E, et al. Hum Mol Genet 2018) and some loci contain more than one causal SNP. Is the opinion of this reviewer that the instrument should be updated accordingly, otherwise they are building a genetic instrument based on a subset of variants that can be difficult to interpret. A potential solution to avoid the inclusion of poor quality variants is to identify proxies for loci in which the lead variant did not pass the quality control. 

The relevance of these findings on T2D call for the need to investigate hard outcomes such as coronary artery disease. 

Reviewer #2: The paper by Rivas and colleagues explores the causal effect of BMI reduction on risk of diabetes across the spectrum of diabetes risk using MR. This is an important question and I found the manuscript to be well-balanced in its methodological approach and discussion. I do have some questions and suggestions for the authors: 

1. The authors constructed an allele score from the weighted sum of 57 alleles using GIANT results (Locke et al.). However, the authors also state "using correlation with BMI in UK Biobank to determine effect directions". Does this mean the direction of effect of (each of) the 57 polymorphisms determined in the UK Biobank for calculation of the allele score? This will certainly inflate the predictiveness of the instrument and bias results. Can the authors explain, and if this is not already the case, use the same direction of effect as in Locke et al.?

2. Why did the authors choose MR-Egger? Can the authors provide results using alternative method (IVW-MR and weighted median, for example)? Was horizontal pleiotropy a concern? If not, other methods are likely to be better powered.

3. A key assumption of the Staley and Burgess method (and all methods looking for non-linear relationships using MR in general) is that the effect of the IV on the exposure is linear and constant for all individuals across the entire of the exposure distribution. Have the authors checked this is the case?

4. It is unclear to me when and how the BMI genome-wide association study the authors have performed in each subset was used in results. If I understand correctly, 57 SNPs from the Locke paper were used as instruments, either with MR-Egger (and results of the diabetes GWAS in each subset) or as a weighted allele score for analysis of nonlinear relationships in 100 BMI quantiles. Can the authors clarify?

5. I suggest adding a column for non-overweight individuals (BMI<25) in Table 1. This would provide an estimate of baseline risk for individuals with a healthy BMI, as compared to "all BMI", which includes overweight and obese individuals.

6. The authors have used a log scale for the y-axis of Figure 1. I wonder if this might mask the rather spectacular effect of BMI on diabetes risk. If felt appropriate, I would suggest using a linear scale.

7. The authors state (page 12) that "However, individuals with a family history of diabetes were predicted to have reduced risk-modification ability through BMI compared to individuals without." This is not readily apparent to me from Table 2. In fact, ORs were consistently larger for individuals with a family history as compared to without. Also, is this difference significant?

8. Figure 2 and Table 2 provide the same information. Would it be possible to combine?

9. I find it difficult to interpret some of the comparisons for predicted weight loss because the "overweight" category includes both overweight and obese individuals. It would be more intuitive to restrict the overweight category to overweight participants (i.e. BMI from 25 to 29.9). To some extent, it is not surprising the two categories have similar risk reduction as they are largely overlapping as currently defined.

10. The labeling of Figure S2 does not match the figure legend.

11. Use of 100 BMI quantiles might obscure any non-linearity because of the large imprecision of estimates in each quantile (figure S3). Can the authors repeat using fewer quantiles?

Reviewer #3: This manuscript uses data from the UK Biobank to examine the causal impact of body mass (BMI)/obesity on type 2 diabetes. They look at prevalence of type 2 diabetes stratified by BMI categories, diabetes family history, and diabetes risk scores. Subsequently, they used Mendelian Randomization across BMI categories to test the reduced T2D risk per kg/m^2 reduction in BMI. Despite increases in T2D prevalence across BMI categories, predicted risk reduction per unit decrease in BMI is roughly comparable ~20-30% decrease across categories. These results are all broadly consistent with previous estimates of the impact of BMI on T2D risk and strongly suggest that weight loss is an effective intervention for diabetes prevention across all weight and genetic/environmental risk categories. The example showing risk reduction for a sample person in terms of weight reduction is useful. 

The paper is clearly written with a straightforward hypothesis. It provides clear examples of why this information is useful and indicates the similar utility of weight loss in T2D prevention for all strata. I have only a few questions/comments.

Comments:

1) My main question regarding the presentation of this analysis is why the results are presented with overlapping BMI categories? I can understand presenting the full distribution results (all BMIs), but it seems the analyses would be more logical if presenting discrete categories (such as typical BMI breakpoints <25, 25-30, >30) that are distinct from each other as opposed to categories that are nested within each other (>25, and >30). 

2) Since many more than 69 SNPs have been associated with BMI, I assume the 69 SNPs used (subsequently pruned to 57) were chosen due to significant association in European-descent samples, correct? Some comment about this specific selection is warranted. 

3) In so far as a noticeable signal of differences exists in these data, it is potentially between those with a family history and those without. Family history encompasses both a shared genetic component and a shared environmental component. Can we use these data to tease apart the two aspects as to which is contributing to the slightly declined value of weight loss in preventing T2D with a family history of T2D? It looks, in comparing odds ratios to T2D PRS to be rather complex. 

4) I am confused by Figure S3. I assume that the lines extending from the point estimates are confidence intervals of some sort (not indicated). Can you explain why the confidence intervals are largest in the center of the BMI distribution where the bulk of the data lies and smaller at the tails? This seems counterintuitive. 

Reviewer #4: I confine my remarks to statistical aspects of this paper. These were very well done and I have no problems recommending publication.

[LINK]

---

## [Decision Letter · Decision Letter 1]

23 Oct 2019

Dear Dr. Rivas,

Thank you very much for re-submitting your manuscript "Homogeneity in the Association of Body Mass Index with Type 2 Diabetes across the UK Biobank: A Mendelian Randomization Study" (PMEDICINE-D-19-02124R1) for review by PLOS Medicine.

I have discussed the paper with my colleagues and the academic editor and it was also seen again by reviewers. I am pleased to say that provided the remaining editorial and production issues are dealt with we are planning to accept the paper for publication in the journal.

[LINK]

We look forward to receiving the revised manuscript by Oct 30 2019 11:59PM. 

Sincerely,

Cathryn Lewis, 

Division of Genetics and Development 

PLOS Medicine

plosmedicine.org

Requests from Editors:

Please add a space before square brackets for references

Introduction first sentence please consider metabolic condition instead of disease 

Methods- we do not require a patient and public involvement section

Page 8 please rephrase “white British UK Biobank Cohort” and please clarify how the researchers know the participants are British. If this information is not available, please remove all references to Nationality. 

Conclusion- please consider rephrasing “unfavourable genetics” to less stigmatising language 

Page 10- please include a space between inverse-variance-weighted and Mendelian Randomization

Page 14 we suggest you edit the third sentence to include “our results show” or similar 

Comments from Reviewers:

Reviewer #2: The authors have satisfactorily addressed all my concerns - thanks!

[LINK]

---

## [Editor Report · Decision Letter 2]

1 Nov 2019

Dear Mr. Rivas, 

On behalf of my colleagues and the academic editor, Dr. Cathryn Lewis, I am delighted to inform you that your manuscript entitled "Homogeneity in the Association of Body Mass Index with Type 2 Diabetes across the UK Biobank: A Mendelian Randomization Study" (PMEDICINE-D-19-02124R2) has been accepted for publication in PLOS Medicine. 

PRODUCTION PROCESS

PRESS

PROFILE INFORMATION

Thank you again for submitting the manuscript to PLOS Medicine. We look forward to publishing it. 

Best wishes, 

Adya Misra

Senior Editor

PLOS Medicine

plosmedicine.org